# Acute seizure activity in neonatal inflammation-sensitized hypoxia-ischemia in mice

Angelina June[1]*, Weronika Matysik[1], Maria Marlicz[1], Emily Zucker[2], Pravin K. Wagley[1], Chia-Yi Kuan[3], Jennifer Burnsed[1,4]

1 Department of Pediatrics, University of Virginia, Charlottesville, Virginia, United States of America,
2 College of Arts and Sciences, University of Virginia, Charlottesville, Virginia, United States of America,
3 Department of Neuroscience, University of Virginia, Charlottesville, Virginia, United States of America,
4 Department of Neurology, University of Virginia, Charlottesville, Virginia, United States of America

* angelina.s.june@gmail.com

## Abstract

### Objective

To examine acute seizure activity and neuronal damage in a neonatal mouse model of inflammation-sensitized hypoxic-ischemic (IS-HI) brain injury utilizing continuous electroencephalography (cEEG) and neurohistology.

### Methods

Neonatal mice were exposed to either IS-HI with *Escherichia coli* lipopolysaccharide (LPS) or HI alone on postnatal (p) day 10 using unilateral carotid artery ligation followed by global hypoxia (n = 10 [5 female, 5 male] for IS-HI, n = 12 [5 female, 7 male] for HI alone). Video cEEG was recorded for the duration of the experiment and analyzed for acute seizure activity and behavior. Brain tissue was stained and scored based on the degree of neuronal injury in the hippocampus, cortex, and thalamus.

### Results

There was no significant difference in acute seizure activity among mice exposed to IS-HI compared to HI with regards to seizure duration (mean = 63 ± 6 seconds for HI vs mean 62 ± 5 seconds for IS-HI, p = 0.57) nor EEG background activity. Mice exposed to IS-HI had significantly more severe neural tissue damage at p30 as measured by neuropathologic scores (mean = 8 ± 1 vs 23 ± 3, p < 0.0001).

### Interpretation

In a neonatal mouse model of IS-HI, there was no significant difference in acute seizure activity among mice exposed to IS-HI compared to HI. Mice exposed to IS-HI did show more severe neuropathologic damage at a later age, which may indicate the presence of chronic inflammatory mechanisms of brain injury distinct from acute seizure activity.

**Data Availability Statement:** The data underlying the results presented in the study are uploaded as supporting information files.

**Funding:** This study was supported by the National Institute of Neurological Disorders and Strokes, National Institutes of Health (NIH) in the form of a grant to JB [K08NS101122] and by the Department of Pediatrics, University of Virginia in the form of a UVA Fellow Grant-in-Aid to AJ [GF003996].

**Competing interests:** The authors have declared that no competing interests exist.

## Introduction

Neonatal hypoxic-ischemic encephalopathy (HIE) affects 1.0 to 8.0 per 1000 live births and is a major cause of neonatal morbidity and mortality worldwide [1]. HIE is also the most common cause of seizures in the neonatal period [1,2]. Risk factors for HIE are multivariate and include sentinel events such as chorioamnionitis [3]. Therapeutic hypothermia (TH) is the current standard-of-care to treat neonatal HIE [4]. However, not all infants with HIE benefit from TH (number needed to treat (NNT) of 7 to decrease mortality or moderate-to-severe neurodevelopmental delay (NDD) at 18 to 24 months) [4]. Factors that may diminish the efficacy of TH include delay in treatment, degree of brain injury, and individual attenuating variables such as underlying inflammation [5–7]. Even with TH, infants who survive HIE remain at high risk for persistent neurodevelopmental deficits, including epilepsy, cerebral palsy, and learning differences [4,8].

Perinatal infections are a common and significant risk factor for neonatal HIE and can worsen outcomes for infants affected by HIE [3,9–11]. Previous studies have shown that the combination of infection and HIE significantly increases the risk of poorer neurodevelopmental outcomes [9–12]. This effect may be due to inflammation-sensitization, a process that occurs when existing multi-organ inflammation in the body leads to increased sensitivity or responsiveness to subsequent inflammatory stimuli [12–14]. In neonatal HIE, inflammation-sensitization may contribute to the development of more severe neurologic injury by increasing the sensitivity of neurons to subsequent injury [15]. Research in this area is ongoing, but there is thought to be a neuroinflammatory synergy between hypoxic-ischemic (HI) injury and inflammation caused by infection [15–17]. In animal studies, *E. coli* LPS is frequently used to model inflammation-sensitized HI (IS-HI) as it activates pathways that produce the same inflammatory cytokines associated with and elevated in HI, including interleukin-6 (IL-6) and tumor necrosis factor-$\alpha$ (TNF-$\alpha$) [13,14,17–19]. These shared inflammatory pathways may underlie the pathophysiology of inflammation-sensitization specific to HI. Both animal and clinical studies have suggested that TH, the only approved therapy for HIE at present, has a reduced or even detrimental effect in IS-HI [20,21]. A more complete understanding of the pathophysiology and long-term sequelae of IS-HI may aid in the development of novel and adjunctive neuroprotective strategies for this at-risk neonatal population.

Seizures in the neonatal period are a common complication in both HIE and severe infection [2,22]. Research is ongoing on the effects of seizure activity in neonatal HIE, including whether seizures independently cause brain injury by increasing local brain metabolism or reflect the underlying evolution of neonatal HI injury [22–24]. In terms of neonatal seizures in the setting in inflammation-sensitization, a neonatal rat study described greater histologic brain injury with lipopolysaccharide (LPS)-induced inflammation in combination with induced status epilepticus [25]. Electroencephalography (EEG) can be utilized with animal models of neonatal HIE, such as the Rice-Vannucci model, to replicate several key findings seen in humans with HIE including clinical and electrographic seizures, abnormal background activity, burst suppression, and status epilepticus [26–28]. A recent study utilized amplitude-integrated EEG (aEEG) to detect electrographic seizures in a piglet model of neonatal IS-HI brain injury in the setting of TH and found aEEG tracing suppression following HI [21,26]. While an excellent screening tool for seizure activity, aEEG is less sensitive and specific in detecting seizures compared to continuous electroencephalography (cEEG) monitoring [27]. No studies have examined acute seizure activity, semiology, and neurohistologic brain injury in an animal model of neonatal IS-HI brain injury utilizing cEEG monitoring.

The goal of this study was to examine acute seizure activity and neuronal damage in a neonatal mouse model of IS-HI brain injury. We hypothesized that IS-HI would cause more

severe acute seizure activity and contribute to greater brain injury compared to HI alone. To test this, we exposed neonatal mice to inflammation-sensitization utilizing *E. coli* LPS, performed cEEG, and analyzed brain tissue with immunofluorescence (IF).

## Materials and methods

### Animals

All animals used for this study were handled according to a University of Virginia Animal Care and Use Committee approved protocol #4071. Housing was in accordance with the National Institutes of Health Guide for Care and Use of Laboratory Animals (US Department of Health and Human Services 85–23, 2011). C57Bl/6 mice (Charles River Laboratories, Wilmington, MA, USA) were used for EEG and histology. Mice of both sexes were used in all experiments (n = 10 [5 female, 5 male] for IS-HI, n = 12 [5 female, 7 male] for HI alone, n = 4 [1 female, 3 male] for sham, n = 6 [3 female, 3 male] for LPS only).

### HI and sham procedure

HI injury was modeled in postnatal day (p) 10 mice with permanent unilateral left carotid artery ligation followed by 1 hour of recovery and then 1 hour of hypoxia at $FiO_2 = 0.08$, as previously described [28–30]. Mice were anesthetized with isoflurane during carotid ligation. The surgical field and hypoxia/EEG recording chamber were situated on warming pads where temperature was controlled to maintain normothermia (36˚C). The p10 mouse is neurologically translatable to the near-term or full-term neonate [28,30,31].

The sham procedure included neck incision and equivalent anesthesia exposure without carotid ligation or hypoxia.

### Inflammation-sensitization

Inflammation-sensitization was performed by injecting *E. coli* LPS as previously described [18]. LPS (0.3 mg/kg #L2630, Sigma-Aldrich, St. Louis, MO, USA) was injected intraperitoneally to p10 mice 4 hours prior to unilateral left carotid artery ligation (Fig 1A). Hypoxia was then induced 1 hour at $FiO_2 = 0.08$ following unilateral left carotid artery ligation. A subset of animals injected with LPS did not undergo HI to control for possible effects of LPS alone on EEG.

### Continuous video EEG

To characterize acute seizure activity and behavior, C57Bl/6 mice had continuous video EEG recording throughout the experimental period as previously described [28,29]. Unipolar insulated stainless steel depth electrodes (0.005 inches bare diameter, 0.008 inches coated; A-M Systems, Sequim, WA, USA) were stereotactically implanted in the bilateral parietal cortex (-1.22 mm Dorsal-Ventral (DV), ± 0.5 mm Medial-Lateral (ML), -1.75 mm Deep (D)) along with reference electrodes in the cerebellum on p9 [29]. Mice were anesthetized with isoflurane during EEG headset implantation. Following 24 hours of recovery, mice were exposed to IS-HI, HI, or sham procedure as previously described. A unity gain impedance matching head stage (TLC2274 Quad Low-Noise Rai-to-Rail Operational Amplifier; Texas Instruments, Dallas, TX, USA) was used for recordings. The baseline EEG recording for animals in the HI group began prior to carotid ligation for 30 minutes, continued through ligation recovery for 1 hour, hypoxia at $FiO_2 = 0.08$ for 1 hour, and reoxygenation for 30 minutes [28,29]. The baseline EEG recording for animals in the IS-HI group began prior to LPS administration for 30 minutes. The pups were subsequently exposed to LPS and then placed back with their mothers

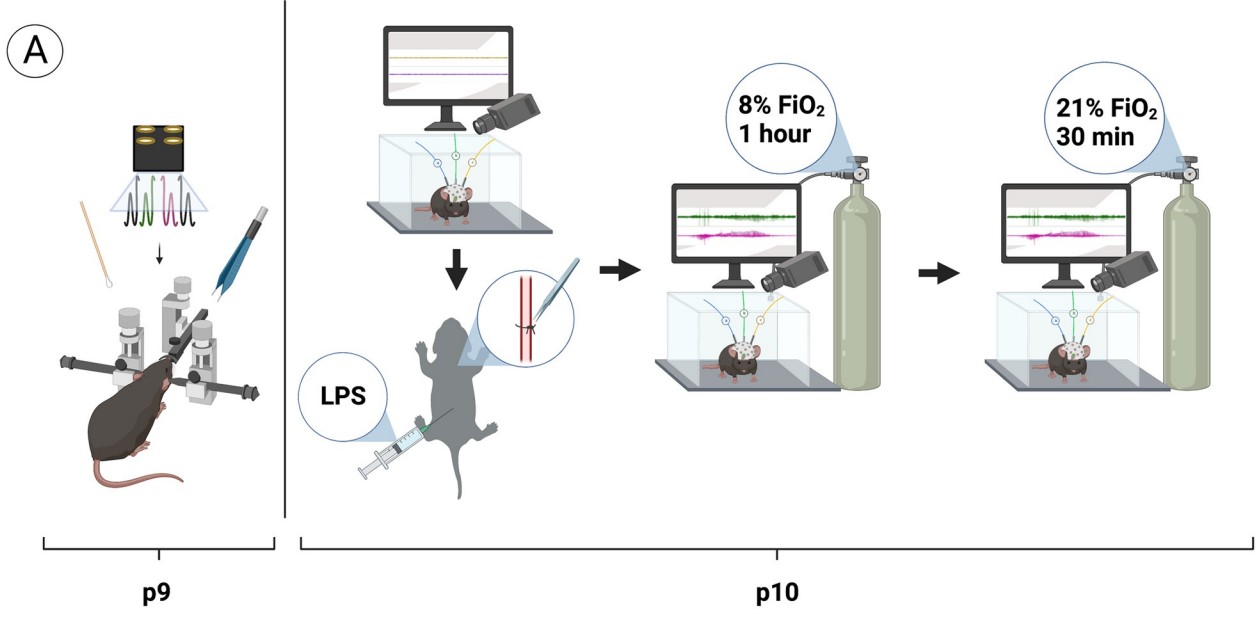

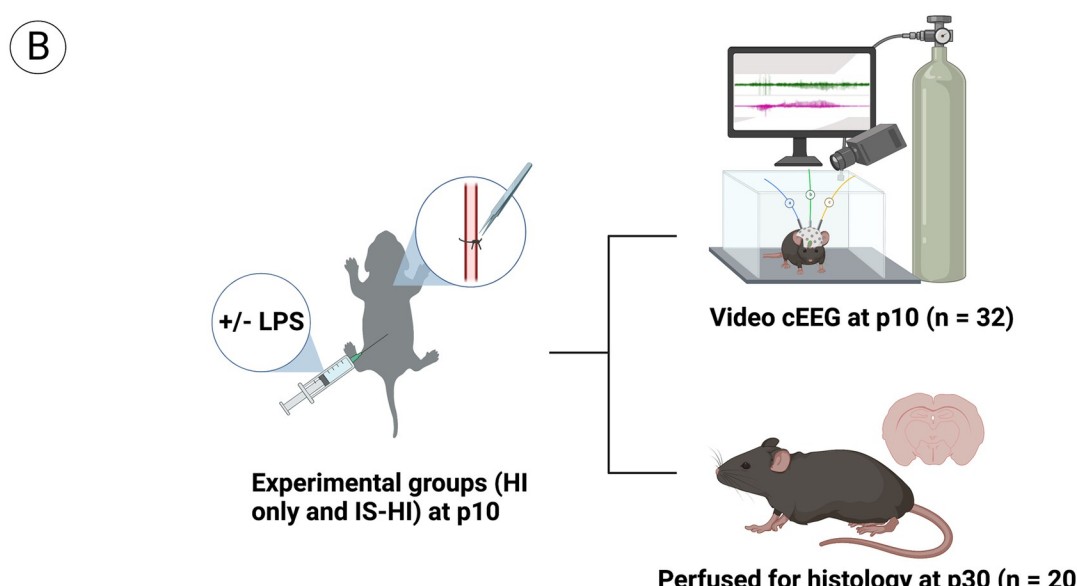

**Fig 1. Experimental design.** (**A**) Experimental timeline and setup of a neonatal mouse model of IS-HI. (**B**) A subset of mice was utilized for brain histology. Created with BioRender.com.

for 4 hours. At 4 hours after LPS injection, unilateral left carotid artery ligation was performed. EEG was reconnected and continued through ligation recovery for 1 hour, hypoxia at $FiO_2 = 0.08$ for 1 hour, and reoxygenation for 30 minutes [28,29]. This equates to a total of 4 hours of video EEG recording for each animal in the HI and IS-HI experimental groups.

### Brain histology and imaging

A subset of animals that did not undergo continuous video EEG monitoring were euthanized at p30 (Fig 1B). Brains from these animals were postfixed, processed, and sliced into sections

for analysis of lesion size in the cortex, hippocampus, and thalamus. Briefly, immunofluorescence (IF) was performed, as previously described, on free-floating 40μm coronal sections [28]. 4',6-diamidino-2-phenylindole (DAPI staining solution; ab228549; Abcam, Cambridge, United Kingdom) staining nuclear DNA was utilized to visualize anatomy and tissue injury in the fixed sections [28].

Imaging was performed using a Zeiss 780 confocal/multiphoton microscope system with Zeiss Zen software for image acquisition (Carl Zeiss, Oberkochen, Germany). For large whole-slice images, 10x magnification was used and, in regions of interest, 20x magnification was used. Excitation wavelengths used for AlexFluor488 was 488nm. Emission filter ranges blue and green were 358 to 461nm and 502 to 550nm, respectively. Tiled images with a Z-stack interval of 10 μm were stitched (overall 15% for 20x images) using Zen software.

## Data analysis

LabChart Pro (ADInstruments, Colorado Springs, CO, USA) was used to collect and analyze video EEG data. Video EEG was reviewed and marked for seizures by a researcher (A.J.) blinded to treatment group and validated with randomly excerpted segments marked by a second blinded researcher (J.B.). Reviews were compared to confirm agreement. Descriptive statistics were performed in GraphPad Prism 9.5.0 (Dotmatics, San Diego, CA, USA) and reported as either mean ± standard deviation or median ± standard deviation. Total number of seizures, total seizure duration, individual seizure duration and length, and behavioral seizure score (BSS) were recorded for each animal [32]. Criteria for electrographic seizure and background attenuation were as previously described [28].

Power spectrograms were derived using LabChart Pro. Background activity was measured by mean voltage extracted from 10-second random excerpts of EEG over the experimental time period. Hypoxia time was divided into "hypoxia 1" and "hypoxia 2," each lasting 30 minutes for a total of 1 hour of hypoxia. Reoxygenation time was also divided into "reoxygenation 1" and "reoxygenation 2" each lasting 15 minutes for a total of 30 minutes of recorded reoxygenation time. Each animal's baseline served as its own control as data following baseline was reported as a percentage of the baseline.

Imaris 10.0 software (Bitplane Scientific, Zurich, Switzerland) was used for colocalization analysis of 5 consecutive 40μm coronal slices per brain (Paxinos and Franklin's The Mouse Brain in Stereotaxic Coordinates, plates 43–47) [33]. Each slice was scored for lesion size and tissue damage by a researcher (W.M.) blinded to treatment group and sex and validated by a second researcher (E.Z. and M.M.) blinded to treatment group and sex, utilizing a modified version of previously described neuropathologic lesion severity scoring scales in this model (Table 1) [14,34,35].

**Table 1. Neuropathologic scoring scale.** The maximum score for each region is 4 (cortex, thalamus, hippocampus). The maximum score for each brain slice, with 3 specific regions analyzed, is 12. The maximum score per animal, with 5 brain slices obtained per animal, is 60. Adapted from Eklind (2001), Hedtjärn (2002), Bona (1998) [14,34,35].

| Brain region | Score | Tissue characteristics |
|---|---|---|
| Cortex Thalamus Hippocampus | 0 | No histopathological damage |
| | 1 | Small infarcts or hypoplasia < or = to 25% compared to right hemisphere |
| | 2 | Medium infarcts or hypoplasia < or = to 50% compared to right hemisphere |
| | 3 | Large infarcts or hypoplasia > 50% compared to right hemisphere |
| | 4 | Total disintegration and/or complete destruction of tissue |

## Results

### Baseline characteristics and survival

To characterize acute seizure activity and behavior in neonatal mice exposed to IS-HI compared to HI alone, electrodes were implanted in the bilateral parietal cortex and video EEG data obtained as described above (n = 10 [5 female, 5 male] for IS-HI and n = 12 [5 female, 7 male] for HI alone). There was also a sham group (n = 4, [1 female, 3 male]) and an LPS injection group that did not undergo HI (n = 6, [3 female, 3 male]). Mortalities of mice included in this analysis include n = 3 [1 female, 2 males] in the IS-HI group, all during hypoxia following a convulsive seizure, and n = 0 in the HI only group. All mice in the sham and LPS only groups survived.

### Acute seizure activity

Preinjury baseline EEG activity was similar to that previously described with no electrographic seizure activity seen [28,29]. Consistent with prior studies, mice in the sham group and mice in the LPS only (no HI) group did not exhibit clinical or electrographic seizures [28]. All mice exposed to IS-HI and HI exhibited electrographic seizure activity. Characteristic EEG patterns were observed in all mice exposed to HI (Fig 2). A total of 184 seizure events were recorded (8 ± 5 events per mouse).

Behavior was characterized with a neonatal rodent BSS [32]. Characteristic seizure semiology generally fell into more severe-range patterns including limb clonus and unstable posture (BSS = 4), limb clonus and unstable posture for > 30 seconds or loss of posture (BSS = 5), and severe tonic-clonic behavior with inability to regain loss of posture (BSS = 6) [32]. Non-convulsive electrographic seizures (BSS = 0–2) were also seen while the pups were motionless and inactive (Fig 3A and 3B). There was no significant difference in median BSS between the IS-HI group and HI group, respectively 5 and 6 (p = 0.25, unpaired t-test).

A subset of mice (n = 5 [3 HI only, 2 IS-HI]) experienced seizures while recovering from carotid artery ligation before being exposed to hypoxia and all were convulsive. Seventy-two percent of mice (n = 16 [8 HI only, 8 IS-HI]) had nonconvulsive seizures which occurred only during hypoxia and reoxygenation. Seizures began on average 16 ± 24 seconds following the start of hypoxia in the HI and IS-HI groups combined. In the reoxygenation phase, thirty-six percent of mice (n = 8 [4 HI only, IS-HI]) continued to have seizures, which trended towards nonconvulsive seizure activity (n = 5 [2 HI only, 3 IS-HI]).

Of the 184 seizure events recorded, each seizure lasted 63 ± 56 seconds. There was no significant difference in seizure duration between the IS-HI and HI group (mean = 63 ± 6 vs 62 ± 5

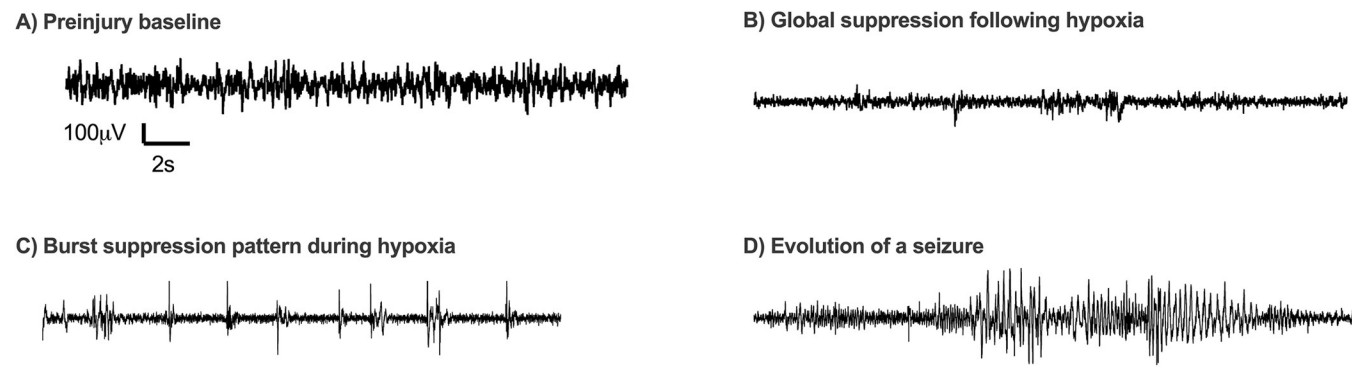

**Fig 2. Characteristic EEG patterns during HI recorded from the parietal cortex depth electrode ipsilateral to ligated carotid artery.** (**A**) Preinjury baseline. (**B**) Global suppression following hypoxia. (**C**) Burst suppression pattern during hypoxia. (**D**) Evolution of a seizure.

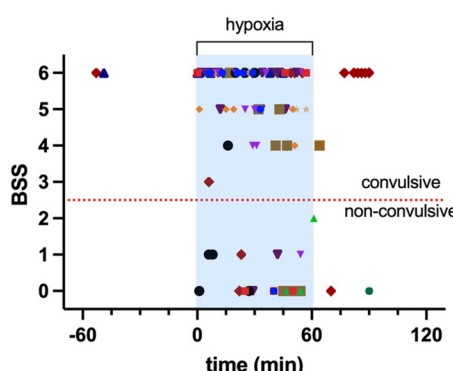

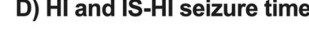

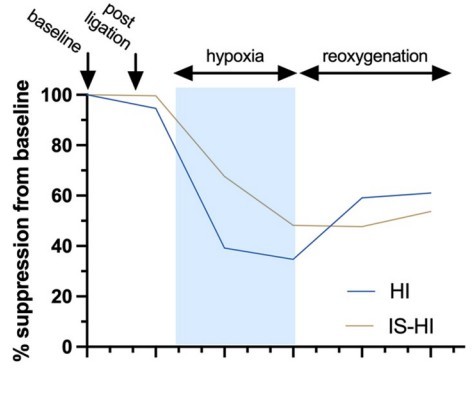

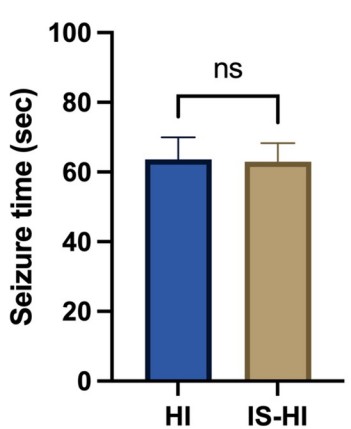

**Fig 3. Seizure behavior, background, and duration in mice exposed to IS-HI vs HI. (A)** BSS and timing for all seizure events in the HI only group (n = 10 mice, each mouse has a unique symbol, each point is a discrete seizure event). **(B)** BSS timing for all seizure events in the IS-HI group (n = 12 mice). **(C)** Background suppression during hypoxia and reoxygenation. Background activity was measured by mean voltage extracted from 10-second random excerpts of EEG over the experimental timeline. Each animal's baseline served as its own control as data following baseline was reported as a percentage of the baseline. Measurements taken from cortical electrodes. **(D)** No significant difference in seizure duration (s) between the IS-HI and HI group (mean = 63 ± 6 vs 62 ± 5 seconds, p = 0.57).

seconds, p = 0.57) (Fig 3D). A subanalysis by sex showed that there were no significant differences in seizure duration between males and females in the IS-HI or HI groups (p = 0.51, two-way ANOVA). Seizure burden, defined here as the number of seizures multiplied by the total seizure time per animal, did not vary significantly between the IS-HI and HI only group (6030 vs 5526 seconds, p = 0.4).

## EEG background activity

As previously described, in both the IS-HI and HI groups, amplitude in both hemispheres progressively decreased following induction of hypoxia relative to baseline activity. In both the IS-HI and HI groups, background amplitude recovers following hypoxia during the reoxygenation phase. The IS-HI group appears to have a slower rate of recovery compared to the HI

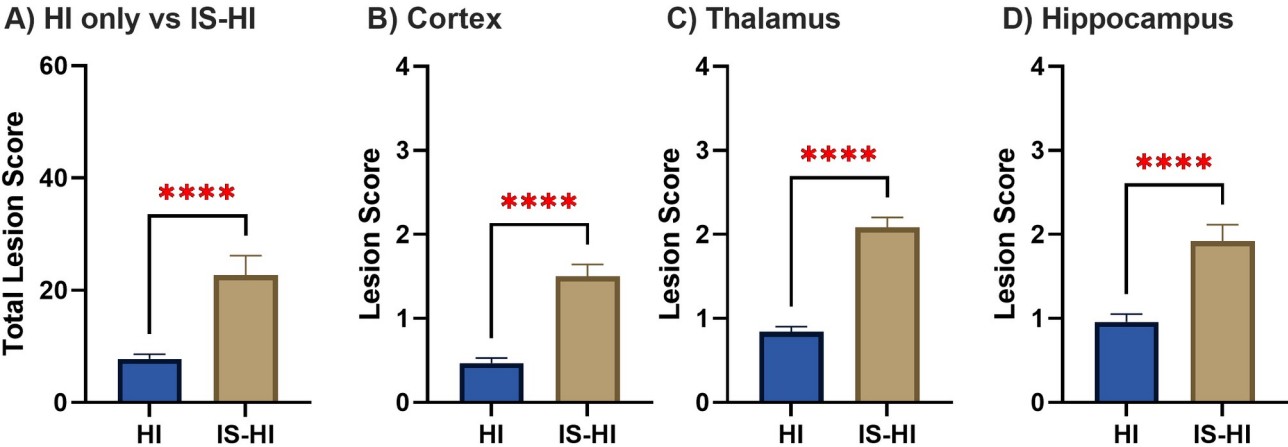

**Fig 4. Neuropathologic scoring. (A)** The IS-HI group (mice exposed to LPS) had significantly higher total neuropathologic scores per animal compared to the HI only group, suggesting more severe neural tissue injury (mean = 8 ± 1 vs 23 ± 3, p < 0.0001). **(B), (C), (D)** The IS-HI group had significantly higher neuropathologic scores in each region of the brain assessed: Cortex (mean = 0.5 ± 0.1 vs 1.5 ± 0.2, p < 0.0001), thalamus (0.8 ± 0.1 vs 2.1 ± 0.1, p < 0.0001), and hippocampus (0.9 ± 0.1 vs 1.9 ± 0.2, p < 0.0001). Data are expressed as mean ± SEM.

only group (Fig 3C). However, the slope of recovery between the IS-HI and HI group do not significantly differ (p = 0.29, simple linear regression) nor does the degree of suppression at each discrete timepoint significantly differ (hypoxia 1 p = 0.05, hypoxia 2 p = 0.18, reoxygenation 1 p = 0.42, reoxygenation 2 p = 0.51, unpaired t-test).

## Brain histology

A subset of mice that did not undergo electrode implantation and continuous video EEG was euthanized at p30 [n = 10 (5 female, 5 male) in the IS-HI group and n = 10 (5 female, 5 male) in the HI only group]. Brains were postfixed and processed into 40μm coronal sections. Lesion size and severity were scored using a modified version of a previously described lesion scoring scale by two blinded researchers (Table 1) [14,34,35].

The mice in the IS-HI group had significantly higher total neuropathologic severity scores compared to the HI only group (Fig 4). A subgroup analysis by sex showed that there was no significant difference in the total neuropathologic severity scores between males and females in the IS-HI group (males mean ± SEM = 26.1 ± 5.15 vs females mean ± SEM = 19.3 ± 4.73, p = 0.17) or the HI only (males mean ± SEM = 6.6 ± 0.85 vs females mean ± SEM = 8.9 ± 1.48, p = 0.09) group.

DAPI was used to visualize anatomy and in slices used for lesion scores. Representative images show lesion size on DAPI on the ipsilateral side of HI injury (Fig 5) [28,36].

## Discussion

The objective of this study was to examine acute seizure activity and neuronal tissue damage in a neonatal mouse model of IS-HI brain injury EEG and neurohistology. Neonates exposed to inflammation-sensitization in the setting of hypoxia-ischemia have increased morbidity and mortality compared to being exposed to hypoxia-ischemia alone. We hypothesized that mice exposed to IS-HI would have greater acute seizure burden compared to HI alone, which may contribute to increased injury in IS-HI, which has been previously described in the literature. However, we found that acute seizure activity in the IS-HI exposed group was similar to that in the HI alone group. Consistent with prior studies, mice exposed to IS-HI exhibited more

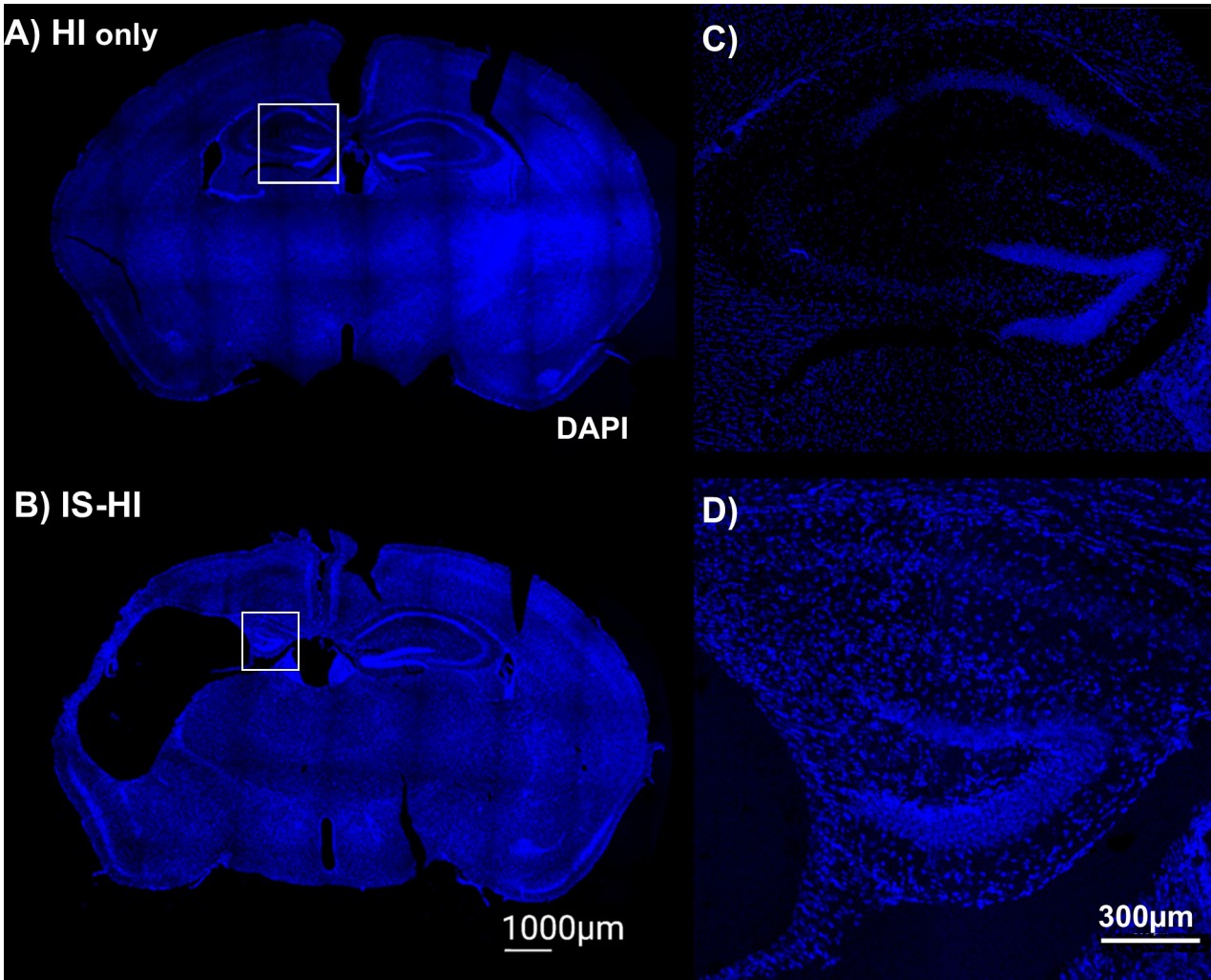

**Fig 5. Representative images of injury in HI only and IS-HI mice.** A subset of mice was euthanized at p30 and brains were postfixed and processed with 40 μm-thick coronal sections and stained using DAPI (blue) as described above. The hemisphere ipsilateral to carotid ligation/ischemia is on the left in all slices, contralateral hemisphere on the right. **(A)** A representative image from a mouse exposed to HI only **(B)** A representative image from a mouse exposed to IS-HI **(C)** Inset from (A) of hippocampal injury in CA1 in HI only mouse, and **(D)** Inset from (A) of hippocampal injury in CA1 in IS-HI mouse.

severe neuropathologic damage remote from the acute injury, which may indicate the presence of chronic inflammatory mechanisms of brain injury distinct from acute seizure activity.

Analysis of EEG background in this study showed prolonged suppression during the re-oxygenation phase which was similar in the IS-HI and HI groups. Inflammation-sensitization has been shown to cause increased brain injury in the setting of neonatal HI [13,14,17–19]. Further studies are needed to investigate whether background suppression is significantly prolonged in the sub-acute and chronic phases of IS-HI injury. EEG background during hypoxia in the IS-HI group was suppressed to a degree similar to HI alone, also supporting the lack of differences in acute injury severity between groups. This may be evidence of a prolonged chronicity of neuronal damage in IS-HI, as opposed to acute injury. Given this, it would be interesting to further utilize EEG to compare the effects of TH on background activity following injury between the two models.

Overall, the mice in the IS-HI group had significantly higher neuropathologic severity scores compared to the HI only group at a timepoint (p30) remote from injury. This may indicate the presence of chronic inflammatory mechanisms of injury distinct from acute seizure activity. The scoring system we utilized employed a gross quantitative measurement of tissue injury and destruction. Further studies may be able to better qualitatively and quantitatively investigate markers of chronic neuronal injury and repair.

TH has been shown to be less neuroprotective in IS-HI compared to HI [20,21]. Proposed mechanisms for this include persistent microglial activation, upregulated production of inflammatory cytokines, and more severe multi-organ dysfunction in the setting of IS-HI [20,21]. Research is ongoing to both better understand why TH is less effective and to find novel neuroprotective therapies in IS-HI. Our study specifically provides data on acute seizure activity in IS-HI, as a biomarker of acute injury, as well as a potential contributing factor in the severity of injury in the acute phase.

In severe HIE, TH may reduce seizure burden and improve neurologic outcomes [19,23]. There is currently variable data on whether seizures in neonatal HIE independently cause brain injury by increasing local brain metabolism or if seizures solely reflect the evolution of HI injury [22–24,37]. However, if TH is less effective in the setting of IS-HI, there may be greater seizure burden in this population overall. Despite the mixed evidence on the effects of seizures in neonatal HIE, aggressive treatment of neonatal seizures is commonplace in clinical practice. The International League Against Epilepsy (ILAE) and World Health Organization published a consensus-based recommendation to use phenobarbital as the first-line anti-seizure medication to treat neonatal seizures regardless of etiology based on a review of 47 studies evaluating first-line treatment of neonatal seizures [22,24,38,39]. In the emerging era of precision medicine, and in the absence of benefit from TH, there may be opportunities to optimize anti-seizure therapy in the setting of IS-HI as an important adjunctive neuroprotective strategy for this uniquely vulnerable population [40]. Further pre-clinical and clinical research to better understand acute seizure activity in IS-HI is crucial to this.

Limitations of this study include the known variability of injury in the Rice-Vannucci model [40,41]. Additionally, rodent models have limited generalizability to humans and may over-simplify the multifactorial etiology of neonatal HIE in clinical practice [41]. *E. coli* LPS specifically models inflammation-sensitization from a Gram-negative infection [14,17,18,21]. Even when using the same IS-HI experimental setup, models of Gram-positive infection can produce different results [21]. Although we followed an established protocol for *E. coli* LPS injection 4 hours prior to HI, the timing of the LPS injection may also affect results [18,42]. In this experiment, histology could not be done on the same animals used for EEG due to permanent headset implantation as the mouse brains used for histology had to be perfused at a later age. Though this was a limitation, it was advantageous to be able to analyze the mouse brains at a later age to shed light on long-term sequelae of IS-HI.

This study provides EEG data on acute seizure activity in neonatal IS-HI. Mice that experience inflammation-sensitized HI do not exhibit more seizures or background suppression on EEG when compared to mice that experience HI only. This indicates that acute seizures do not contribute to the increased severity of injury in IS-HI compared to HI, which has been previously published and demonstrated here. This may indicate that the factors contributing to increased injury in IS-HI are more chronic in nature and not significantly contributing to the acute phase of HI injury. Our pre-clinical data provides potential insights and opportunities to further investigate novel neuroprotective strategies for a population of neonates with IS-HI who may not benefit from the current standard-of-care, TH [20,21]. Further research of the role of acute seizures and chronic inflammation in IS-HI is critical to developing novel, adjunctive, and individualized neuroprotective therapies.

## Supporting information

**S1 File. Acute seizure activity EEG review.**
(XLSX)

**S2 File. Background EEG review and analysis.**
(XLSX)

**S3 File. Neuropathologic injury scores.**
(XLSX)

## Acknowledgments

We thank Dr J. Kapur for his guidance.

## Author Contributions

**Conceptualization:** Angelina June, Pravin K. Wagley, Chia-Yi Kuan.

**Data curation:** Angelina June, Pravin K. Wagley, Jennifer Burnsed.

**Formal analysis:** Angelina June, Pravin K. Wagley, Jennifer Burnsed.

**Funding acquisition:** Angelina June, Jennifer Burnsed.

**Investigation:** Angelina June, Weronika Matysik, Maria Marlicz, Emily Zucker, Pravin K. Wagley, Jennifer Burnsed.

**Methodology:** Angelina June, Pravin K. Wagley, Jennifer Burnsed.

**Resources:** Angelina June, Jennifer Burnsed.

**Supervision:** Angelina June, Pravin K. Wagley, Chia-Yi Kuan, Jennifer Burnsed.

**Validation:** Angelina June, Weronika Matysik, Maria Marlicz, Emily Zucker, Pravin K. Wagley, Jennifer Burnsed.

**Visualization:** Angelina June, Jennifer Burnsed.

**Writing – original draft:** Angelina June.

**Writing – review & editing:** Angelina June, Weronika Matysik, Maria Marlicz, Emily Zucker, Pravin K. Wagley, Chia-Yi Kuan, Jennifer Burnsed.

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
