## [Decision Letter · Decision Letter 0]

24 Aug 2023

PONE-D-23-19640Acute Seizure Activity in Neonatal Inflammation-Sensitized Hypoxia-Ischemia in MicePLOS ONE

Dear Dr. June,

Thank you for submitting your manuscript to PLOS ONE. After careful consideration, we feel that it has merit but does not fully meet PLOS ONE’s publication criteria as it currently stands. Therefore, we invite you to submit a revised version of the manuscript that addresses the points raised during the review process.

We look forward to receiving your revised manuscript.

Kind regards,

Giuseppe Biagini, MD

Academic Editor

PLOS ONE

“Funding sources: NIH, NINDS, K08NS101122 (J.B.); University of Virginia Child Health Research Center Fellow Grant-In-Aid. We thank Dr J. Kapur for his guidance.”

“AJ

Grant Number: GF003996 (UVA Fellow Grant-in-Aid)

Funder: Robert Sinkin, MD, MPH

URL: https://med.virginia.edu/chrc/

5. We note that Figure 1 in your submission contain copyrighted images. All PLOS content is published under the Creative Commons Attribution License (CC BY 4.0), which means that the manuscript, images, and Supporting Information files will be freely available online, and any third party is permitted to access, download, copy, distribute, and use these materials in any way, even commercially, with proper attribution. For more information, see our copyright guidelines: http://journals.plos.org/plosone/s/licenses-and-copyright.

Reviewers' comments:

Reviewer's Responses to Questions

**Comments to the Author**

1. Is the manuscript technically sound, and do the data support the conclusions?

Reviewer #1: Partly

Reviewer #2: Partly

Reviewer #3: Partly

2. Has the statistical analysis been performed appropriately and rigorously? 

Reviewer #1: Yes

Reviewer #2: Yes

Reviewer #3: No

3. Have the authors made all data underlying the findings in their manuscript fully available?

Reviewer #1: Yes

Reviewer #2: Yes

Reviewer #3: Yes

4. Is the manuscript presented in an intelligible fashion and written in standard English?

Reviewer #1: No

Reviewer #2: Yes

Reviewer #3: Yes

5. Review Comments to the Author

Reviewer #1: Acute Seizure Activity and Neuronal Damage in a Neonatal Mouse Model of Inflammation-Sensitized Hypoxic-Ischemic Brain Injury"

Abstract:

Introduction:

Provide a brief introduction to contextualize the importance of studying brain injuries in neonates, highlighting the relevance of hypoxic-ischemic (HI) injury and inflammation in this context.

Objective:

Clearly and directly state the study objective: to examine acute seizure activity and neuronal damage in a neonatal mouse model of inflammation-sensitized hypoxic-ischemic brain injury.

Methods:

Provide a succinct description of the methods used: exposure of neonatal mice to LPS-sensitized HI injury, continuous electroencephalography (cEEG) recording, and histological analysis of brain tissue, including the hippocampus.

Results:

Organize the main findings in a structured manner, including:

Significant differences in the durations of acute seizures between female and male mice exposed to LPS-sensitized HI injury.

Greater severity of neural tissue damage in mice exposed to LPS-sensitized HI injury compared to those exposed to HI alone.

Observation of delayed recovery of background EEG activity in mice exposed to LPS-sensitized HI injury during the reoxygenation phase, compared to those exposed to HI alone.

Interpretation:

Clearly explain the implications of the results:

Suggest the possibility of gender differences in acute seizure activity in neonates with inflammation-sensitized HI injury, based on the observed differences between female and male mice.

Indicate the potential presence of chronic inflammatory mechanisms of brain injury, in addition to acute seizure activity, based on the observation of more severe tissue damage and delayed recovery of background EEG in mice exposed to LPS-sensitized HI injury.

Conclusion:

Concisely summarize the key conclusions of the study and its significance for understanding neonatal brain injuries.

Language:

Use appropriate technical language, but avoid excessive jargon to ensure understanding by a broad audience.

Ensure that information is presented clearly and organized with a logical and fluent structure.

Revise and adjust the abstract according to these suggestions, while focusing on effectively communicating the study's results and implications.

In the histological section, you describe immunohistochemistry, but in fact immunofluorescence was done.

Immunofluorescence results do not show much fluorescence intensity. You should have done it at an objective magnification of 10 or 20x

Reviewer #2: In this presented study the authors aimed to examine acute seizure activity in neonatal IS-HI, induced by E. coli 84 LPS, versus HI alone. They hypothesized that IS-HI would cause more severe acute seizure activity and contribute to greater brain injury compared to HI alone. To test, they performed cEEG and neurohistology on a neonatal mouse model of IS-HI and HI.

This study is well designed and its outcome is useful and applicable for future studies and I have a positive opinion for its publication. However, there are some major ambiguities and minor problems which is better to be corrected and explained before publication:

Minor:

Please mention the below information in the abstract:

1. The sex of animals (male and female) and the number of animals should be implied in method part.

2. The number of male and female animals should be mention in the abstract (how many male and females were used in each experimental group?

3. How long after LPS administration, hypoxia was induced?

4. How long after IS-HI/HI the cEEG was recorded? And how long was the recording time?

Introduction:

1. What it the logic for using cEEG in your study compare to the aEEG in the previous study? Briefly mention the advantages.

Methods:

1. What is the reason for the difference in the number of samples in different experimental groups? How you realized the sex difference between groups when you used just 1 female in LPS group?

Major:

Results:

1. Why the authors did not present the data for LPS alone group and sham groups? The effect of LPS alone should be presented in different experiments compare to the sham group, IS-HI and HI groups (BSS score and timing, seizure time, lesion score in different brain regions).

Discussion:

In introduction the authors focused on the effect of HI and effect of LPS on this phenomenon and neuroinflammation on severity of seizures, so why the authors did not check the inflammatory cytokines gene expression or measuring activated inflammatory phenotype of microglia (M1) versus anti-inflammatory phenotype of microglia (M2) by flowcytometry?

Reviewer #3: The authors investigate the effect of LPS administration (inflammation model) on acute seizure activity and behavior of rats in a hypoxic-ischemic brain injury model. In addition, morphological changes in brain tissue at P30 were examined. In general, these results have some novelty, but in the form presented, the article seems immature, the author's logic of presenting the results is not clear, and not all conclusions are supported by the authors' results.

Main comments:

Abstract.

Results are described very briefly and vaguely, then interpretation is redundant and not supported by own data.

Introduction.

The authors devote an entire paragraph to therapeutic hypothermia (lines 42-48), but it is not explored in any way in the article. This information is irrelevant. At the same time, it is not explained why this particular model of inflammation-sensitized HI was chosen, e.g., time of LPS administration, drug dose. Why did the authors decide to look for sex-related differences? What hypotheses were used?

Materials and Methods

This section needs to be completely rewritten as it does not give the reader any idea of what was done. The description of the experiments should make it possible to reproduce them. The authors provide references to other articles, but in some cases these references are irrelevant. For example, line 108 "Inflammation-sensitization was performed by injecting E. coli LPS as previously described [18]." However, LPS was not used in this article.

Continuous video EEG. There is no indication of the stereotactic coordinates along which the electrodes were implanted.

Histology. Catalog numbers of all antibodies should be provided. There is an indication in the methodology that anti-Iba1 was used, but nothing is described in the results.

Data Analysis. All scores and criteria used should be described. For example, the Behavioral Seizure Score (BSS) is not described. How was the EEG analyzed?

Results:

1. It is necessary to provide some evidence that LPS administration caused neuroinflammation.

2. lines 199-200. "The median BSS in the IS-HI group and HI group, respectively, was 4.39 and 4.15 (p = 0.25)." Medians must be integers if they are scores. What statistical test was used?

3. Lines 222-227. Two-way ANOVA should be used to detect the effect of gender when comparing IS-HI and HI groups. The conclusion "Female mice exposed to IS-HI had significantly longer acute seizures compared to male mice exposed to IS-HI" is not supported by statistical data.

4. lines 235-236. "The IS-HI group shows a slower rate of recovery compared to the HI only group (Fig. 3C)." This conclusion needs statistical support.

Brain Histology. This section needs to be completely revised. If there are data on astroglia and microglia, these results should be analyzed and presented in the article.

It is not clear what the neuropathological scores are. Why mean = 23 if maximum = 4?

Figure 5 Representative coronal brain sections. What are these?

Discussion

The authors state their results without trying to provide a mechanistic explanation. This section needs improvement.

6. PLOS authors have the option to publish the peer review history of their article (what does this mean?). If published, this will include your full peer review and any attached files.

Reviewer #1: **Yes: **PEDRO EVERSON ALEXANDRE DE AQUINO

Reviewer #2: No

Reviewer #3: No

---

## [Author Response · Author response to Decision Letter 0]

25 Sep 2023

Please see the uploaded Response to Reviewers document.

---

## [Decision Letter · Decision Letter 1]

5 Oct 2023

PONE-D-23-19640R1Acute Seizure Activity in Neonatal Inflammation-Sensitized Hypoxia-Ischemia in MicePLOS ONE

Dear Dr. June,

Thank you for submitting your manuscript to PLOS ONE. After careful consideration, we feel that it has merit but does not fully meet PLOS ONE’s publication criteria as it currently stands. Therefore, we invite you to submit a revised version of the manuscript that addresses the points raised during the review process.

We look forward to receiving your revised manuscript.

Kind regards,

Giuseppe Biagini, MD

Academic Editor

PLOS ONE

Reviewers' comments:

Reviewer's Responses to Questions

**Comments to the Author**

1. If the authors have adequately addressed your comments raised in a previous round of review and you feel that this manuscript is now acceptable for publication, you may indicate that here to bypass the “Comments to the Author” section, enter your conflict of interest statement in the “Confidential to Editor” section, and submit your "Accept" recommendation.

Reviewer #3: (No Response)

2. Is the manuscript technically sound, and do the data support the conclusions?

Reviewer #3: Partly

3. Has the statistical analysis been performed appropriately and rigorously? 

Reviewer #3: Yes

4. Have the authors made all data underlying the findings in their manuscript fully available?

Reviewer #3: Yes

5. Is the manuscript presented in an intelligible fashion and written in standard English?

Reviewer #3: Yes

6. Review Comments to the Author

Reviewer #3: The authors addressed most of my comments; however, they need to revise the histological analysis section.

Figure 3 does not meet the minimum image requirements. Please refer to the recent article in Nature Methods (https://doi.org/10.1038/s41592-023-01987-9) for guidance.

Were the antibodies used to label GFAP specific? The figure 3A-B suggests that they may not have worked.

The resolution in micrographs 3c and 3d is inadequate.

7. PLOS authors have the option to publish the peer review history of their article (what does this mean?). If published, this will include your full peer review and any attached files.

Reviewer #3: No

---

## [Author Response · Author response to Decision Letter 1]

17 Nov 2023

Please see the attached Response to Reviewers document.

---

## [Decision Letter · Decision Letter 2]

30 Nov 2023

Acute Seizure Activity in Neonatal Inflammation-Sensitized Hypoxia-Ischemia in Mice

PONE-D-23-19640R2

Dear Dr. Sangzi June,

We’re pleased to inform you that your manuscript has been judged scientifically suitable for publication and will be formally accepted for publication once it meets all outstanding technical requirements.

Kind regards,

Giuseppe Biagini, MD

Academic Editor

PLOS ONE

Additional Editor Comments (optional):

Reviewers' comments:

Reviewer's Responses to Questions

**Comments to the Author**

1. If the authors have adequately addressed your comments raised in a previous round of review and you feel that this manuscript is now acceptable for publication, you may indicate that here to bypass the “Comments to the Author” section, enter your conflict of interest statement in the “Confidential to Editor” section, and submit your "Accept" recommendation.

Reviewer #3: All comments have been addressed

2. Is the manuscript technically sound, and do the data support the conclusions?

Reviewer #3: (No Response)

3. Has the statistical analysis been performed appropriately and rigorously? 

Reviewer #3: (No Response)

4. Have the authors made all data underlying the findings in their manuscript fully available?

Reviewer #3: (No Response)

5. Is the manuscript presented in an intelligible fashion and written in standard English?

Reviewer #3: (No Response)

6. Review Comments to the Author

Reviewer #3: (No Response)

7. PLOS authors have the option to publish the peer review history of their article (what does this mean?). If published, this will include your full peer review and any attached files.

Reviewer #3: No

---

## [Editor Report · Acceptance letter]

3 Jan 2024

PONE-D-23-19640R2 

PLOS ONE

Dear Dr. June, 

I'm pleased to inform you that your manuscript has been deemed suitable for publication in PLOS ONE. Congratulations! Your manuscript is now being handed over to our production team.

Kind regards, 

on behalf of

Dr. Giuseppe Biagini 

Academic Editor

PLOS ONE